# Multi-Scenario Simulation of Land Use and Landscape Ecological Risk Response Based on Planning Control

**DOI:** 10.3390/ijerph192114289

**Published:** 2022-11-01

**Authors:** Nan Wang, Peijuan Zhu, Guohua Zhou, Xudong Xing, Yong Zhang

**Affiliations:** 1School of Geographical Sciences, Hunan Normal University, Changsha 410081, China; 2Hunan Key Laboratory of Land and Resources Evaluation and Utilization, Changsha 410007, China; 3Hunan Key Laboratory of Geospatial Big Data Mining and Application, Hunan Provincial Normal University, Changsha 410081, China; 4Hunan Sidayuan Planning Consulting Research Co., Ltd., Changsha 410081, China

**Keywords:** planning control, land use, multi-scenario simulation, landscape ecological risk

## Abstract

This study applied territorial spatial planning control to a land use multi-scenario simulation in Changde, China, and measured the landscape ecological risk response. It embedded five planning control schemes, respectively, involving inertial development, urban expansion size quantity control, ecological spatial structure control, land use zoning control, and comprehensive control. Findings show that: (1) Woodland and arable land in Changde occupy 31.10% and 43.35% of land use, respectively, and constitute the main functional space of the research area. The scale of construction land in Changde has enlarged continuously, with ecological space represented by woodland and water constantly squeezed and occupied. (2) Comprehensive control has the most remarkable restraining effect on the disordered spread of construction land, while ecological space structure control is the most effective way to control ecological land shrinkage. (3) The overall landscape ecological risk index expanded over 2009–2018, presenting an S-type time evolution curve of “sharp increase–mitigation”. Landscape ecological risk presents a single-core, double-layer circle structure with the north and east regions as the core, attenuating to the periphery. (4) Landscape ecological risk under land use zoning control increased significantly more than in other scenarios. Comprehensive control best prevented landscape ecological risk and restrained the disorderly expansion of construction land.

## 1. Introduction

Rapid global urbanization and continued economic and population growth have forced dramatic changes in land use [1]. The high-speed economic development model, supported by a large number of natural resources, makes the spatial allocation of land development inclined to production and construction. Urban space has expanded in a disorderly way to meet the demands of extensive urban land use development brought about by rapid urbanization. Structural conflicts and governance conflicts between the ecological environment and land use development have intensified [2]. The risk of non-linear changes in ecosystems has increased and has resulted in a series of problems, such as huge pressure on the ecological environment and the “fragmentation” of ecological patches [3,4,5]. Regional ecological security has been constantly challenged by events such as natural disasters, degradation of ecological functions, and imbalances of ecosystems, which seriously restrict the sustainable, high-quality development of the regional social economy. Therefore, a sustainable urban development plan that resolves the contradiction between the ecological environment and land use development is a hot topic of global concern [6]. Countries around the world are actively exploring planning schemes to avoid ecological security risk brought about by land use changes, including Planetary Boundaries [7,8], ecological security patterns [9], urban growth boundaries (UGB) [10], and green infrastructure (GI) [11]. Among them, China has promulgated “Several Opinions on Establishing a Territorial Spatial Planning System and Supervising the Implementation” of it. The governance orientation of the coordination of territorial space development and protection has gradually become clear. Restricting the inefficient and disorderly development of land by means of planning control is an important spatial governance method to resolve the “zero-sum game” between ecological environmental protection and land use development.

Among the main effects of human activities on the environment are land use and resulting land cover changes [12]. Land use and cover change (LUCC) was proposed as a joint initiative of the Humanities Field Program (IHBP) and the International Geosphere Biosphere Program (IGBP) at the 1995 Global Environmental Change Conference [13]. LUCC has gradually become a research hotspot and front-line issue against the background of current global environmental changes and sustainable development [14,15]. LUCC is a nonlinear change process that is highly compounded and impacted by multi-factor subsystems, such as natural ecology, the economy, society, and policy formulation and implementation. It can directly represent the external performance of mutual feedback between the natural environment and human activities, which can lead to abrupt changes in the spatiotemporal characteristics of ecological subsystems, such as aquatic ecology [16,17,18], atmospheric environment [19], and species [20]. The ecological impact of LUCC will lead to changes in the spatial and temporal characteristics of the ecological landscape pattern, resulting in ecological risks [21,22,23], such as ecological degradation and soil dehydration. Thus, territorial spatial planning control is important to optimize the interaction between the natural environment and land use and to resolve the conflict between social and economic development and ecological protection.

The concept of territorial spatial planning control comes from land use control. It refers to scientific planning and arrangement for various development and protection activities in land space. It is also an important policy tool for macro-control governance and micro-fine management of land use. Humans develop and protect land space through management and control of it, thereby changing the orientation of land use and ensuring ecological security [24]. Current research is mainly focused on the interaction mechanism between LUCC and ecosystems under global environmental change [25,26] and the extrapolation and simulation prediction of LUCC trends at different geographic scales [27,28,29], and there are few studies on the relationship between planning control, LUCC, and ecological security. Land use prediction simulation models can be classified into three categories according to their predictive properties [30]: (1) Non-spatial simulation models use mathematical models as prototypes to predict and simulate the overall scale parameters of land use. (2) Spatial simulation models use the location integration model to simulate the future development direction of LUCC. (3) Coupling models couple multiple models on the basis of the spatial simulation model and integrate the mathematical model and location model to simulate and predict the evolution process of complex land use systems (Table 1). However, their application still lacks a deep understanding of territorial space planning control and management efficiency. Therefore, it is difficult to form effective guidance for spatial governance under the real situation of territorial spatial planning. Relevant fields for predicting future land use change under different control scenarios need to be enriched. There are few studies on the relationship between planning control, LUCC, and ecological security.

Understanding and quantifying LUCCs and their ecosystem responses under planning control is critical to understanding the relationship between human social development and the natural environment. Carrying out LUCC simulation research to reveal the relationship between territorial spatial planning control and LUCC law can inform the alleviation of the contradiction between man and land. Therefore, this study described the temporal and spatial differentiation characteristics of land use evolution from 2009 to 2018 with Changde in Hunan Province in China, as the research area, and set up scenario conversion rules for multiple planning and control schemes, which were built into the Future Land Use Simulation Model (FLUS). This study attempted to explore the difference in characteristics and evolution rules of land use simulation results under different planning control scenarios, and carried out landscape ecological risk measurement across different scenarios, so as to describe the spatiotemporal response and feedback characteristics of landscape ecological risk under different land use change directions.

The aim of this study was to provide a research framework to evaluate the effectiveness of territorial spatial planning and control implementation. The framework combined a land use multi-scenario simulation model with a landscape ecological risk model. It was designed as an analytical tool to further characterize the management and control effectiveness of different types of territorial spatial planning and thereby provide theoretical reference and practical experience in order to improve the efficiency and quality of land development and avoid landscape ecological risk, as well as to help planners more scientifically and rationally judge and evaluate the management efficiency of planning management and control methods for the development direction of land use and provide a path to optimize the management and control methods of territorial space planning.

## 2. Study Area and Materials

### 2.1. Study Area

The city of Changde (110°29′–112°18′ E, 28°24′–30°07′ N), located in the northwest of Hunan Province, is the scope of this study. Its total land area is 18,178.25 km^2^. In 2019, the resident population was 5.772 million, and the urbanization rate of the resident population was 54.5%. It has jurisdiction over nine districts and counties (cities) and six management areas (Figure 1).

Changde is an important area for undertaking the radiation of Chang-Zhu-Tan and promoting industrial development. Changes in land use in Changde have been significant. At the same time, the contradiction between land space development and ecological protection has continued to intensify, and the landscape’s ecological pattern has also been severely disturbed. Therefore, this study selected Changde as the study area for its typicality or representativeness.

### 2.2. Materials

The data required for this study are shown in Table 2 according to the research content and method data requirements, combined with the feasibility of data acquisition.

## 3. Methods

In this study, we proposed a research framework to assess the effectiveness of territorial spatial planning and control implementation. The framework is the integration of the FLUS model and the landscape ecological risk model. This framework can be used to assess and predict the effectiveness of the management and control of territorial spatial planning on a regional scale. The overall structure of the research framework is shown in Figure 2.

### 3.1. Markov Model

The Markov model is a LUCC simulation tool suitable for describing changes [39]. The Markov model describes LUCC from one period to another and applies it to predict future changes. This is achieved by developing a LUCC random transition matrix from time one to time two. The transfer matrix represents the probability of land conversion from one category to another. For example, three types of land result in nine possible use changes. It determines the nature of changes and is the basis for predicting future periods [40]. The Markov model is widely used to forecast the demand scale of land use development in future land use simulations [41]. One of the advantages of the Markov model is its simplicity. It can describe the complex and long-term land use conversion process with simple transfer probability, which is one of the most effective methods to estimate the transition of each land use type because it uses a convenient and effective calculation method, namely the transition random matrix.

The application of the Markov model in this study included two parts: first, we used the land use data of Changde in 2009 and 2015 to forecast the demand scale of each land use type in 2018, so as to simulate and forecast the land use change in 2018. We also verified the accuracy with the actual situation in 2018 to ensure that the FLUS model has high reliability. Second, we used the land use data of Changde in 2009 and 2018 to obtain the land use transfer probability matrix of the study area in 2030. We adjusted the transfer probability matrix according to different scenario rule settings so as to adapt to the difference in scenarios to obtain different land use demand scales. We simulated the land use change in 2030 under different scenarios. The specific calculation formula is as follows:(1)$$S_{t+1}=P_{ij}\times S_{t},$$
(2)$$P_{ij}=\left[\begin{array}{ccc} P_{11} & \cdots & P_{1n}\\ \vdots & \ddots & \vdots \\ P_{n1} & \cdots & P_{nn} \end{array}\right]\;P_{ij}\in [0,\;1)\;\;and\;\sum_{n=1}^{n}P_{ij}=1\left(i,\;j=1,\;2,\ldots ,n\right)$$
where, $S_{t+1}$, $S_{t}$ is the distribution state of land use types at time t+1 and time t, respectively; $P_{ij}$ represents the probability of converting land use type i to land use type j.

### 3.2. The FLUS Model

The FLUS model is a method that uses an artificial neural network (ANN) model algorithm to perform iterative and repeated training simulations to quantitatively measure the possibility of land use type conversion [34,42]. The existing FLUS model consists of two parts: one is to combine the neural network algorithm with the land use data, and consider the natural, social, economic, and other driving factors to obtain the adaptability probability within the scope of land type research. The second is to combine the suitability probability obtained with the land use grid data and match the corresponding conversion rules so as to better avoid error accumulation. Research in the existing literature shows that the FLUS model can be better applied to future land use scenario simulations and can play a strong role in spatial layout optimization, land use simulation, and other auxiliary decision-making.

This study used GeoSOS-FLUS to simulate future land use change in Changde, which mainly included two parts:1Calculating the suitability probability of land use types according to the basic land use data and driving factor data set of Changde in 2009. After verifying that the root mean square error (RMSE) of suitability probability meets the accuracy requirements, we input the 2018 land use data and driving factor data on GeoSOS-FLUS software to calculate the 2018 land use suitability probability, providing a basis for the simulation of the FLUS model in 2030.2Simulating and predicting land use in the study area in 2030 using the CA adaptive inertial mechanism. Designing various parameters of the FLUS model, such as restricted conversion area, domain factor, and inertial coefficient, and input the Markov model cost transfer matrix obtained from 2009 and 2018 land use data into the model. In the iterative simulation process, the overall conversion probability of each cell is evaluated, and finally, the land use simulation in the study area is realized.

#### 3.2.1. ANN-Based Suitability Probability Estimation

The FLUS model simulates the probability and spatial distribution of different land use types using historical land use change data as the initial data and the driving factor data of land use change using the method of the artificial neural network. The neural network is the abstraction of the human brain’s neural network. It has strong self-organization, self-learning habits, automatic control, and other characteristics. It can generate high probability distribution suitability and establish the relationship between occurrence probability and driving factors of various land types through optimization calculation of the complex geographic system. Each neuron in ANN corresponds to a variable in CA and is composed of an input layer, a hidden layer, and an output layer.

The simulation process of the neural network is to establish the spatial relationship between the initial land use type and the driving factors [43]. The specific calculation formula is as follows:
(3)$$p\left(p,k,t\right)=\sum_{j}w_{j,k}\times sigmoid\left(net_{j}\left(p,t\right)\right),$$
(4)$$sigmoid\left(net_{j}\left(p,t\right)\right)=\frac{1}{1+e^{-net_{j}\left(p,t\right)}},$$
(5)$$net_{j}\left(p,t\right)=\sum^{\;}w_{i,j}\times c,$$
where p(p,k,t) represents the suitability probability of land use type k at grid p time t, and the sum of the suitability probabilities of all land use types is 1; $w_{j,k}$ represents the weight between the hidden layer j and the output layer k; $sigmoid\left(net_{j}\left(p,t\right)\right)$ represents the correlation function from the hidden layer to the output layer; $net_{j}\left(p,t\right)$ is the signal received by the grid p at t time in the hidden layer j; and $w_{i,j}$ represents the weight between the input layer i and the hidden layer j.

#### 3.2.2. Land Use Scenario Simulation Based on Self-Adaptive Inertia and Competition Mechanism CA

In the FLUS model, we input design parameters, such as the suitability probability distribution and transfer cost matrices of various land use types, limited transfer area, domain factor, and inertia coefficient, etc. Then, we evaluate the overall transition probability for each cell to achieve the best output fit for each type of cell. Finally, we simulate the land use situation in the study area.

1Weight of neighborhood: Weight of neighborhood refers to the spatial interaction factor between different cells. In this study, the weight of a neighborhood can be characterized as the expansion ability of a certain land use type. It is calculated as follows:

(6)$$\Omega_{p,k}^{t}=\frac{\sum_{N\times N}con\left(c_{p}^{t-1}=k\right)}{N\times N-1}\times w_{k},$$
where ${Ω}_{p,k}^{t}$ represents the weight of the neighborhood, that is, the probability that a land use type k may appear in cell p at iteration time t; $\underset{N\times N}{\sum }con\left(c_{p}^{t-1}=k\right)$ is the total number of cells occupied by the k-th land use type at the latest iteration time t−1 in the window of *N* × *N*; and $w_{k}$ represents the variable weight of domain factors among different land use types.

2Inertia coefficient: The inertia coefficient is to improve the continuity of the land use type in the cell by changing its inertia coefficient, so as to change the development trend of land use. It is calculated as follows:

(7)$$Inertia_{k}^{t}=\{\begin{array}{c} Inertia_{k}^{t-1}\;if\left|D_{k}^{t-1}\right|\leq \left|D_{k}^{t-2}\right|\\ Inertia_{k}^{t-1}\times \frac{D_{k}^{t-2}}{D_{k}^{t-1}}\;ifD_{k}^{t-1}<D_{k}^{t-2}<0\\ Inertia_{k}^{t-1}\times \frac{D_{k}^{t-1}}{D_{k}^{t-2}}\;if0<D_{k}^{t-2}<D_{k}^{t-1} \end{array},$$
where $Inertia_{k}^{t}$ represents the inertia coefficient of land use type k at iteration time t, and $D_{k}^{t-1}$ represents the difference between land demand and the actual quantity of land use type k at iteration time t−1.

3Comprehensive probability: Comprehensive probability is defined as the functional relationship between adaptive probability, the neighborhood factor, and the inertia coefficient. It is calculated as follows:

(8)$$TP_{p,k}^{t}=p\left(p,k,t\right)\times {Ω}_{p,k}^{t}\times {Inertia}_{k}^{t}\times \left(1-SC_{c\to k}\right),$$
where, $TP_{p,k}^{t}$ is the comprehensive probability that a land use type k may appear on cell *p* at iteration time t, and $SC_{c\to k}$ indicates the conversion energy consumed or the conversion cost paid for the conversion from land use type c to land use type k.

### 3.3. Model Validation

This study simulated land use change in 2018 on the basis of predicting the scale of land use demand and the setting of the corresponding model parameters, and compared the simulation result with the actual situation. The study calculated the Kappa coefficient and overall accuracy of the simulation to verify the validity of the simulation model. The value interval of the Kappa coefficient and overall accuracy is [0, 1]. The closer the value of the Kappa coefficient and overall precision are to 1, the closer the simulated data are to the spatial pattern of the actual data. When the Kappa coefficient is greater than or equal to 0.8, it indicates that the simulation accuracy has reached a good level. Kappa is calculated as follows:(9)$$K=\frac{N\sum_{i=1}^{r}x_{ii}-\sum_{i=1}^{r}\left(x_{i+}\times x_{+i}\right)}{N^{2}-\sum_{i=1}^{r}\left(x_{i+}\times x_{+i}\right)},$$
where, r is the number of rows and columns of the confusion matrix, that is, the number of different LU categories used to assemble the confusion matrix. In this study, the LU category is 6, so r=6; $x_{ii}$ represents the number of pixels correctly simulated; $x_{i+}$ represents the total number of pixels in row i; $x_{+i}$ represents the total number of pixels in column i; and N is the total number of pixels.

### 3.4. Landscape Ecological Risk Model

Ecological risk refers to the possibility and potential hazard degree of adverse impact on the assessment endpoint caused by risk factors such as environmental pollution, human stress, or natural disasters in different types of ecosystems on a regional scale. In order to explore the risk coercion of land use change on the ecosystem, this study introduced a more specific concept of landscape ecological risk. Landscape ecological risk refers to the corresponding landscape pattern disturbed by natural or human factors on a regional scale. This study measured landscape ecological risk based on the framework of landscape fragmentation—landscape separation—landscape dominance, and emphasized the territorial spatial planning control of the whole area, all types, and all elements. Among them, the landscape fragmentation index can describe the morphology of landscape types at the patch level. The landscape separation index is a comprehensive reflection of the interweaving and connection of material flow, energy flow, and information flow within the overall landscape pattern. The landscape dominance index is a representation of the ecological importance of different landscape types in the landscape pattern. The measurement of landscape ecological risk in this study focused more on the comprehensive characterization of the overall pattern of landscape ecology. On the one hand, it could better describe the pressure response characteristics of the overall land space caused by external disturbances such as land use evolution; on the other hand, it could also characterize the vulnerability heterogeneity of the internal landscape of the land space. The evaluation indicators and models are shown in Table 3.

The objective assignment method can objectively determine its weight through the information entropy of the statistical data, thereby eliminating the limitations of the subjective assignment method [43]. Therefore, this study used the CRITIC weighting method on the SPSS AU platform to weigh the indicators, as shown in Table 4.

This study was different from the traditional way of assessing landscape ecological risk by land use type classification. On the one hand, this study integrated woodland, water, grassland, wasteland, and other land use types with landscape ecological functions into ecological land, so as to fully reflect the relationship between landscape structural characteristics and landscape ecological risk. On the other hand, the land use types were divided into three types of landscape types: construction land, agricultural land, and ecological land to carry out landscape ecological risk measurement so as to meet the goals of territorial spatial planning control. The landscape vulnerability index of the three landscape types is shown in Table 5.

## 4. Results

### 4.1. Analysis of Land Use Evolution and Analog Verification

#### 4.1.1. Analysis of Land Use Evolution

The evolution characteristics of land use structure are shown in Table 6. Woodland and arable land were the main land use types in Changde, and the two together accounted for more than 70% of the total area. Woodland and water were evidently being squeezed, with a net decrease of 84.88 km^2^ and 64.52 km^2^ from 2009 to 2018. Construction land expanded significantly, from 1519.70 km^2^ in 2009 to 1617.64 km^2^ in 2018. The rapid advancement of urbanization has led to a strong demand for the expansion of construction land, which would inevitably squeeze woodlands and other ecological spaces. The arable land showed a steady growth trend, with a net increase of 68.48 km^2^, an increase of 38%.

This study used the overlay analysis tool of ArcGIS to establish the land use transition matrix (Figure 3). In general, the transfer of construction land was the hotspot of the land use transfer trend in 2009–2018. The transfer out of ecological land, such as woodlands and water, was the most severe, indicating that the ecological construction pattern has not yet stabilized in some areas. Arable land, construction land, and wasteland were the main types of transfer, and the transfer-in areas were 68.36 km^2^, 97.93 km^2^, and 3.23 km^2^. Woodland, grassland, and water were the main types of transfer-out, and the transfer-out areas were 85.17 km^2^, 20.23 km^2^, and 64.91 km^2^.

#### 4.1.2. Analog Verification of Land Use Evolution

This study selected Changde land use data in 2009 as the original layer. We imported the data of eight driving factors into the neural network fitness probability calculation module and set the relevant parameters. Finally, we obtained the results of the suitability probability distribution of Changde (Figure 4), and its root mean square error (RMSE) was 0.27, indicating that its land use data training accuracy was high.

This study verified the accuracy of the simulation results. The overall accuracy was 0.93 and the Kappa coefficient was 0.90, both of which have met the accuracy requirements, indicating that the simulation results had a high fit and could more accurately and truly reflect the future evolution of land use. The prediction simulation applied to the future land use development had high reliability. This study could be used to predict the spatial growth of land use in Changde in 2030.

### 4.2. Multi-Scenario Simulation and Analysis of Land Use Change Based on Planning Control

#### 4.2.1. Multi-Scenario Settings Based on Planning Control

This study sorted out and summarized the current status of territorial space planning and control in Changde. It could be divided into urban expansion size quantity control, ecological spatial structure control, and land use zoning control according to control methods. The urban expansion size quantity control is mainly based on the municipal index system constructed under the requirements of the Guidelines for Compilation of Municipal Territorial Space Overall Planning (Trial Implementation). The ecological spatial structure control is to maintain the ecological security pattern under the dual constraints of the Changde ecological corridor control line and the ecological protection red line. The land use zoning control was to divide and designate the first-level basic zoning oriented by territorial spatial planning and formulate corresponding zoning control rules based on principle guidance, bottom-line control, and directory control.

1Inertia Development Scenario

The inertial development scenario did not consider the binding impact of any planning control scheme on land use change, and the land use demand still continued the current land use change trend for urban expansion. Therefore, we directly predicted land use demand by the Markov model based on land use data in 2009 and 2018, and we did not set a restriction on conversion area in the GeoSOS-FLUS software.

2Urban Expansion Size Quantity Control Scenario

The urban expansion size quantity control is to control the scale, efficiency, and quality of the land space with the help of various binding indicators of territorial spatial planning so as to rationally allocate and reasonably utilize the national land space. It is a quantitative expression of the final effect of planning, and an important means of territorial space control. According to the Changde City Land Utilization Master Plan (2016–2020) index system, the total scale of urban construction land should be controlled within 1805.53 km^2^, and the amount of arable land should not be less than 5801.35 km^2^. Therefore, we set the land use scale in GeoSOS-FLUS software accordingly.

3Ecological Space Structure Control Scenario

The ecological space structure control is to ensure the transmission of material flow, energy flow, and information flow in ecological space and the function of maintaining biodiversity through ecological corridor control line constraints. A sufficient buffer space should be formed around the ecological corridor, and strict control measures should be implemented within the buffer range of the control line. According to the upper limit of the recommended buffer width of ecological corridors in relevant studies, this study set the buffer width of ecological corridors to 1000 m so as to control the conversion of ecological functional land use types such as woodland, grassland, and water to construction land. Therefore, we reduced the probability of ecological land transfer to construction land by 50% and set an ecological corridor buffer zone and an ecological source area as the land use restricted transformation areas in GeoSOS-FLUS software.

4Land Use Zoning Control Scenario

Land use zoning control is mainly based on the basic zoning control rules, strictly in accordance with the requirements of territorial spatial planning control, and implements land differentiated control. It strictly controls the expansion of construction land to ecological land. Therefore, we reduced the transfer probability of arable land to construction land (40%), woodland (20%), grassland (20%), and water (20%), and finally obtained the corresponding land use demand.

5Comprehensive Control Scenario

Comprehensive control mainly includes three control means: comprehensive urban expansion size quantity control, ecological spatial structure control, and land use zoning control, so as to meet the requirements for territorial spatial planning control. Therefore, this study reduced the transfer probability of arable land to construction land (40%), woodland (20%), grassland (20%), and water (20%), and compared the results with the land use demand of the urban expansion size quantity control scenario to obtain the corresponding land use demand. Then, this study set up the restriction conversion area, including an ecological corridor buffer zone, an ecological source area, an ecological protection zone, an arable land protection zone, and an urban development area.

#### 4.2.2. Analysis of Land Use Simulation Results under Different Scenarios

The simulation results still continued the trend of land use change from 2009 to 2018 because the land use change trend was not constrained by planning control methods. The land use types in Changde were still dominated by woodland and arable land in 2030. The main evolution direction was the expansion of construction land and the reduction of ecological land. In contrast, the expansion trend of construction land had moderated, and the shrinking trend of ecological land was slowing down (Table 7).

In the Inertia Development Scenario, the expansion of construction land was spatially manifested as the extensional expansion mode concentrated in the periphery of towns and the connotative filling mode of construction land in the central urban area. The woodland, grassland, and other ecological land in the central and eastern regions had a significant shrinking trend, showing the characteristics of point shrinkage and edge shrinkage (Figure 5b).

In the Urban Expansion Size Quantity Control Scenario, the expansion of construction land was a typical extensional expansion mode. The urbanization development zone, with Dingcheng District and Wuling District as the centers, showed a more dramatic expansion of construction land. In contrast, the expansion of construction land that was far away from the centralized construction land appeared to be more restrained. The reduction of ecological land was more profound at the intersection of construction land, arable land, and ecological land (Figure 5c).

In the Ecological Space Structure Control Scenario, the expansion trend of construction land to ecological land has been restricted, the stability of the ecological functional space structure has been enhanced, and its reduction trend has converged. In contrast, the amount of land use expansion that was closely related to human development and construction activities has shrunk. The transfer-out effect of woodland and water that are greatly disturbed by human activities has effectively been contained in 2018–2030 (Figure 5d).

In the Land Use Zoning Control Scenario, the marginal expansion trend of construction land was strictly controlled. Building land in central urban areas was more a matter of shrewd growth in space filling. The scattered construction land in the peripheral areas maintained the growth mode of marginal expansion. Ecological land reduction areas were no longer concentrated at the intersection of urban space and ecological space but instead at the intersection of agriculture and forestry (Figure 5e).

In the Land Use Zoning Control Scenario, the construction land showed obvious fragmentation and expansion characteristics in space. The overall shrinking trend of ecological land has moderated. The main body of occupation of woodland and water was transferred from construction land to arable land (Figure 5f).

Table 7 shows the characteristics of land use structures in different scenarios in 2030. The continuous reduction of ecological land and the rapid spread of construction land were the main manifestations of land use evolution. Compared with the inertia development scenario, the scale of construction land expansion in the urban expansion size and quantity control scenario was significantly curbed, and the expansion model was mainly based on extensional development. The trend of construction land spread in the land use zoning scenario has been greatly reduced, the connotative expansion of the concentrated construction land area has become the main expansion mode, and the scattered construction land in the periphery presented characteristics of an enclave-type spread. The shrinking trend of ecological land in the ecological spatial structure control scenario has been effectively curbed, indicating that ecological spatial structure control has absolute advantages in optimizing the ecological spatial pattern and alleviating the fragmentation of ecological patches.

Compared with the inertial development scenario, the degree of inhibition of construction land expansion was in the order of comprehensive control scenario > land use zoning control scenario > ecological spatial structure control scenario > urban expansion size quantity control scenario. This shows that the index constraint on the expansion trend of construction land was still an important means of land use regulation and management in the future, but it lacked the requirements for the guidance and optimization of the territorial spatial pattern. Compared with the inertial development scenario, the order of the degree of ecological land shrinkage mitigation was ecological spatial structure control scenario > comprehensive control scenario > urban expansion size quantity control scenario > land use zoning control scenario. This showed that the ecological spatial structure control could guide and organize ecological space based on the systematicity, integrity, and connectivity of ecological space and effectively restrain the encroachment and destruction of ecological space by human activities. The protection of ecological space by land use zoning control tended to be “mosaic” management and control through the ecological protection red line, and its ecological protection efficiency was inevitably inferior to ecological space structure control.

This study analyzed the overall characteristics of land use distribution and its spatial expansion direction under different scenarios with the help of standard deviation ellipse analysis tools (Figure 6). The five control scenarios could effectively curb the westward and northward expansion directions of construction land. The offset distance of construction land in the comprehensive control scenario was the smallest, and that in the inertial development scenario was the largest. It shows that comprehensive control had the greatest driving influence on the expansion of construction land. The ecological land reduction offset distance in the urban expansion size quantity control scenario was the largest, and that in the ecological space structure control scenario was the smallest. This shows that the ecological space structure control was the most significant for the reduction and driving of ecological land.

### 4.3. Assessment and Analysis of Landscape Ecological Risk

#### 4.3.1. Unit Division of Landscape Ecological Risk

This study used a regular grid to sample land use data to measure the spatial heterogeneity of landscape ecological risk. A 3 km × 3 km grid was selected to divide the landscape ecological risk unit, using the ArcGIS Fishnet tool. Thereby, this study obtained 2214 landscape ecological risk units, and it calculated the relevant landscape risk indicators in the grid, as shown in Figure 1.

#### 4.3.2. Landscape Ecological Risk Assessment Based on Land Use Evolution

1Analysis of the Overall Evolution of Landscape Ecological Risk

The PD and IJI of construction land showed a significant downward trend, and the landscape fragmentation index had decreased. The PD and IJI of ecological land were in a significant upward development trend, and the landscape fragmentation index had increased. The landscape separation index of construction land was the largest, and the ecological land was the smallest, which indicated that the ecological land patches were most concentrated. The SPLIT of construction land continued to decrease significantly, while CONNECT and COHESION increased. This showed that the expansion pattern of construction land patches presented agglomeration and connectivity. The COHESION of ecological land did not change much, while the SPLIT and CONNECT showed a continuous downward trend, indicating that the patch layout of ecological land was more dispersed. The landscape dominance index of construction land was the smallest, and ecological land was the largest. This suggested that the regional landscape was dominated by ecological land. The landscape dominance index of construction land showed a continuous upward trend from 2009 to 2018, which fully reflected the trend of expanding and contiguous construction land. In contrast, the landscape dominance index of ecological land showed a significant downward trend. Table 8 showed that the overall landscape ecological risk index in the study area showed a trend of continuous expansion, and its growth rate also expanded.

2Grid-based Analysis of Spatial and Temporal Differentiation of Landscape Ecological Risk

This study used the ordinary kriging method in the geostatistical method module of ArcGIS 10.2 to interpolate the landscape ecological risk unit space in order to deeply describe the spatial pattern characteristics of landscape ecological risk. To conveniently analyze the evaluation characteristics of the ecological risk index during different periods in the study area, the ecological risk index values of subareas were divided into five grades by using natural break point and a relative index method according to the actual situation of the study area [44,45]. Risk was divided into low-risk areas (ERI ≤ 0.2), medium-to-low-risk areas (0.2 < ERI ≤ 0.3), medium-risk areas (0.3 < ERI ≤ 0.4), medium-to-high risk areas (0.4 < ERI ≤ 0.5), and high-risk areas (ERI > 0.5). Finally, this study obtained the spatial distribution of landscape ecological risk in Changde from 2009 to 2018.

Table 9 showed the change in landscape ecological risk level, which presented an “S”-shaped time evolution curve of “surge–moderation”. Overall, landscape ecological risk increased dramatically. The period from 2009 to 2012 was a period of rapid growth in landscape ecological risk. After 2012, the rising trend of landscape ecological risk moderated. Overall, it showed a trend of increasing at first and then gradually easing.

Figure 7 shows the spatial distribution of ecological risk levels. The landscape ecological risk level showed an obvious trend of decaying from the northern and eastern regions to the periphery, showing a single-core, double-layered structure. The low-risk area and medium-to-low-risk area were mainly distributed in the key ecological function areas in the mountainous areas of the southwest and northwest, scattered in the northwest corner of Shimen County and the southwest corner of Taoyuan County in 2015 and later. The medium-risk area was the middle area where the low-risk area transitioned to the high-risk area, and it was dominated by a staggered area of agricultural land, ecological land, and construction land. The medium-to-high-risk area was mainly distributed in Li County, Anxiang County, Hanshou County, Dingcheng District, and other areas where agricultural land and ecological land were interlaced with construction land. The high-risk area spread from point-like areas in 2009 to Li County, Anxiang County, and Dingcheng District in 2018.

#### 4.3.3. Landscape Ecological Risk Assessment Based on Scenario Simulation Results

1Analysis on the Overall Evolution of Landscape Ecological Risk

This study measured the landscape ecological risk index under five different scenarios and compared it horizontally in order to describe the characteristics of landscape ecological risk aversion ability under different planning control methods.

The results showed that the landscape fragmentation index of construction land was the highest in the urban expansion size quantity control scenario; the landscape fragmentation index of ecological land in the ecological space structure control and comprehensive control scenarios was relatively low; the landscape separation index of construction land in the urban expansion size quantity control scenario and the ecological space structure control scenario was the lowest, and that in the comprehensive control scenario was the highest; the landscape separation index of ecological land in the comprehensive control scenario was the lowest, while that in the urban expansion quantity was the highest; the landscape dominance index of construction land in the urban expansion size quantity control scenario was the highest; and the landscape dominance index of ecological land in the land use zoning control scenario was the lowest (Table 10).

2Grid-based Analysis of Spatial and Temporal Differentiation of Landscape Ecological Risk

The spatial interpolation of landscape ecological risk units in different scenarios was carried out using the geostatistical method module. The landscape ecological risk was divided into five levels according to the spatial clustering method. The results are as follows.

Figure 8 shows the spatial distribution of landscape ecological risk levels in different scenarios. Compared with the inertial development scenario, the reduction in the level of high-risk areas in the urban expansion size quantity control scenario was mainly concentrated in the plain area of Anxiang County in the northeast, and the expansion of low-risk areas was mainly distributed in the mountainous area of Shimen County in the northwest. The expansion of the low-risk area in the ecological space structure control scenario was mainly concentrated in the mountainous areas of Shimen County and Taoyuan County. The reduction of medium-risk areas in the land use zoning control scenario did not lead to the expansion of low- and medium-to-low-risk areas, but the large-scale expansion of medium-to-high- and high-risk areas in Anxiang County, Li County, and other areas in the east. The low-risk area in the comprehensive control scenario expanded from the northern part of Shimen County to most of the county, and the range of medium-to-high-risk was significantly reduced in Li County, Linli County, Anxiang County, and other regions.

Comprehensive control can most effectively reduce landscape ecological risk, followed by ecological spatial structure control. Land use zoning control is weak in the management and control of landscape ecological risk. Urban expansion size quantity control has certain advantages in curbing the fragmented development of construction land and regulating and managing the spread of construction land. Ecological space structure control is very effective for the control of ecological connectivity and system integrity, and can effectively alleviate the fragmentation and separation trend of ecological space. However, it is relatively weak in shaping the shape of construction land and in differentiated management and control. Land use zoning control has the disadvantages of mismatch between production and living activities and ecological space, and “one size fits all” between ecological space structure and production and living functions. The “overcorrection” of land use zoning control not only reduces the flexibility of construction land development, but also increases the risk of fragmentation and separation of ecological land and landscape. Comprehensive control can effectively restrain the inefficient expansion of construction land and optimize the land use layout structure. It can realize the elastic guidance of the ecological space structure and the bottom line control of the ecological protection area. It is the most effective in mitigating the deterioration of landscape ecological risk.

## 5. Discussion

### 5.1. Multi-Scenario Simulation of Land Use and Landscape Ecological Risk Response Based on Planning Control

Previous studies have mainly focused on deducing future land use development based on the law of land use evolution and have set relatively simple constraints. The existing land use prediction and simulation applications still lack a deep understanding of territorial spatial planning control efficiency, and it is difficult to form effective guidance for spatial governance under the real situation of territorial spatial planning. Relevant fields for predicting future land use change under different control scenarios need to be enriched. Since China’s economic reform and opening-up, a high-speed economic development model supported by the consumption of a large number of natural resources has caused a series of problems, such as the spread of inefficient urban land use and huge pressure on the ecological environment. Economic development and ecological protection are caught in the difficult dilemma of a “zero-sum game”. In the face of such a practical dilemma, territorial spatial planning control is an important starting point for the coordination of planning, protection, and development in the new era of ecological civilization. Compared with previous studies, this study proposed a multi-scenario type and parameter design scheme based on planning and control. Land use scenario simulation based on territorial spatial planning control can interpret the development of land use in different planning control methods, thereby providing a theoretical reference to inform policy guidelines for the scientific management and control of territorial space.

### 5.2. Key Points

1Multi-scenario simulation and comparison based on planning control is adopted to make up for the inadequacy of the traditional model in the multi-scenario setting and the insufficient docking of territorial space control methods.2In the constraint scenario, the impact of ecological spatial structure control on land use under the background of ecological civilization is considered, so as to more objectively and reasonably predict the future land use scale under the new territorial spatial planning control system.3In the ecological space structure control, the width of the buffer range of the ecological corridor was delimited in combination with the state’s layout of the construction of nature reserves and major projects for the protection of wild animals and plants. It is of great significance for coordinating the relationship between resource environmental protection and urban development and construction.

### 5.3. Limitations and Future Prospects

1The systematic quantitative research on landscape ecological risk assessment cannot be carried out comprehensively considering all factors due to the diversity, complexity, and nonlinearity of ecosystem dynamics [46,47]. With the deepening of the understanding of the regional ecosystem in the future, a more scientific evaluation model would be constructed to evaluate the landscape ecological risk more systematically and comprehensively.2Some parameters were mainly subjectively set under the guidance of experience in the parameter setting process of scenario simulation. On the one hand, the subjective judgment of the changes of driving factors in the land use system was not scientific and rigorous enough. On the other hand, the inertia of land use development was ignored. Therefore, there was a certain subjectivity in the land use scenario simulation.3The territorial spatial planning control at the municipal level focuses on the control of upper-level constraints and the control of the overall pattern and makes strategic and systematic arrangements for space protection and development. It is recommended to build a planning control transmission system of “quantity + structure + zoning” in the future. Quantity control and transmission reflect the rigid constraints of territorial space and should be further continued and optimized. Structural control and transmission should realize the overall planning of the control of core space elements, and the municipal level should strengthen overall planning and coordinate contradictions. Zoning control and transmission should reflect spatially differentiated control (Figure 9).

## 6. Conclusions

This study designed scenario conversion rules according to the characteristics of multiple planning control methods: inertial development, urban expansion size and quantity control, ecological spatial structure control, land use zoning control, and comprehensive control, built into the FLUS model. It then carried out a multi-scenario simulation of land use based on planning control, with Changde as the case study area, and measured the landscape ecological risk response of different simulations. The conclusions are as follows:

First, the study showed that woodland and arable land still occupied a high land use dominance and were the main functional spaces in the study area. Changde initially formed a land use landscape pattern with woodland and arable land as the matrix, water as corridors, and construction land as patches. The scale of construction land in Changde continued to expand, and the ecological space was in a state of being constantly squeezed and occupied from 2009 to 2018.

The continuous reduction of ecological land and the expansion of construction land were the main manifestations of land use evolution in five scenarios. The expansion of construction land showed the aggregation law of “small area agglomeration and large area dispersion”. The development potential field of future construction in the study area would focus on the value reconstruction and urban renewal of the existing construction land stock space. In the comprehensive control scenario, the effect of curbing the disorderly spread of construction land was the most prominent, and its agglomeration law was the most significant. Different planning control means were relatively difficult to reverse the trend of ecological land reduction. The ecological space structure control was the most effective in regulating the shrinking trend of ecological land. And the encroachment of construction land and arable land on ecological corridors and ecological source land was obviously restrained.

Landscape ecological risk presented an “S”-shaped time evolution curve of “surge-moderation”. The northern and eastern regions had a significant trend of attenuation towards the periphery, showing a single-nuclear, double-layered, and layered structure. The medium-to-high- and high-risk areas were concentrated in the staggered areas of agricultural land and ecological land in Li County, Anxiang County, and Jin City, as well as areas with severe conflicts between construction land and ecological land in the periphery of townships.

The increase in landscape ecological risk in the land use zoning control scenario was significantly higher than in the other scenarios, while the landscape ecological risk in the comprehensive control scenario was smaller than in the other scenarios. This showed that comprehensive control had the effect of reversing risk expansion in the local space. Therefore, comprehensive control was the best for effectively preventing landscape ecological risk and curbing the disorderly expansion of construction land. Ecological spatial structure control could effectively maintain the structural connectivity and integrity of the regional ecosystem, and urban expansion size quantity control had an obvious curbing effect on the expansion of construction land.

It is recommended to consider optimal management and control and minimize negative externalities of landscape ecological risk. Territorial space planning control should implement mandatory quantity control requirements and basic land use zoning of land space at the municipal level. In addition, the physical boundaries of ecological landscape patches should be connected in series with the structural framework of ecological corridors so as to achieve optimal planning and control of land space.

## Figures and Tables

**Figure 1 ijerph-19-14289-f001:**
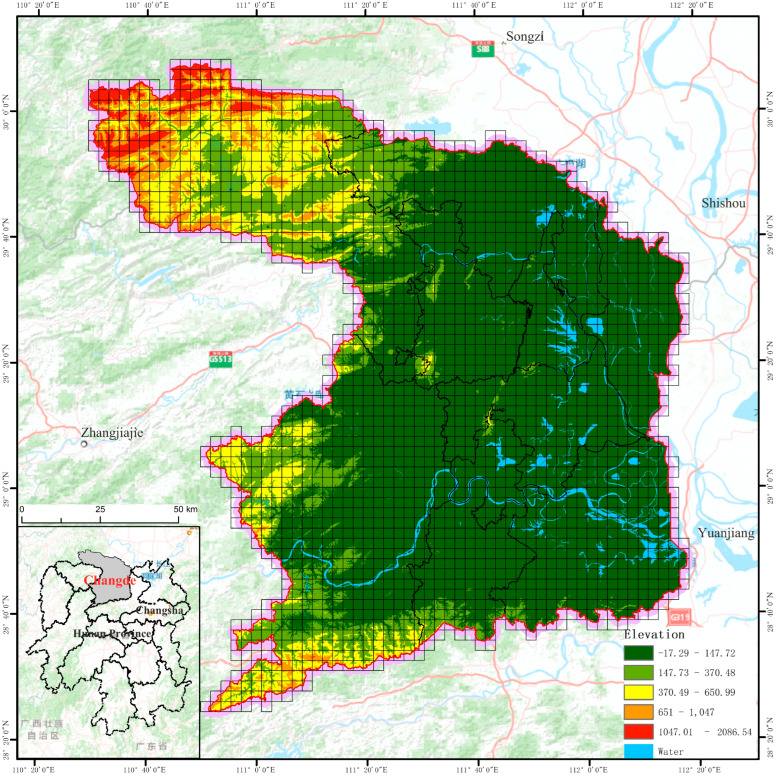
The Spatial Location of the Study Area.

**Figure 2 ijerph-19-14289-f002:**
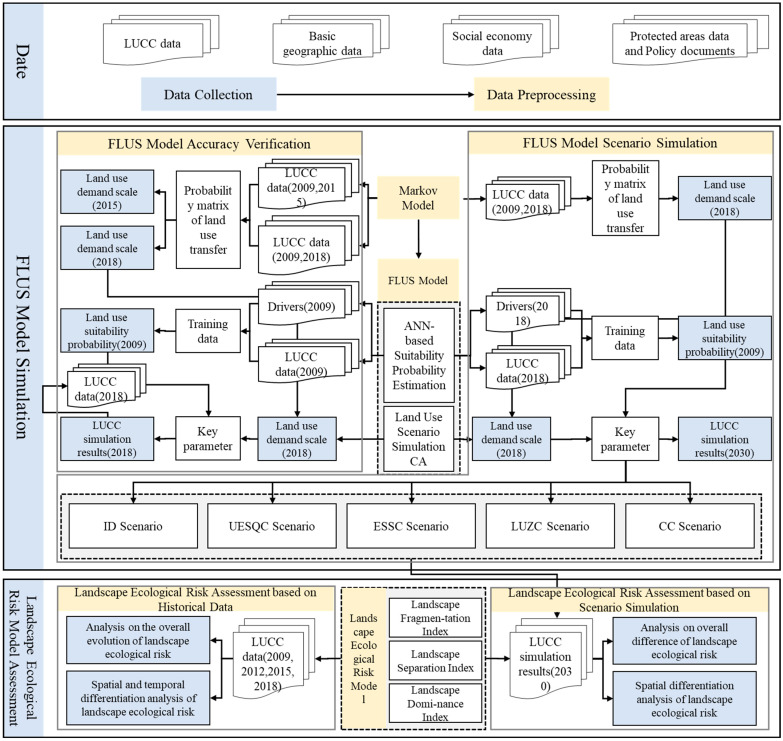
Framework of This Study.

**Figure 3 ijerph-19-14289-f003:**
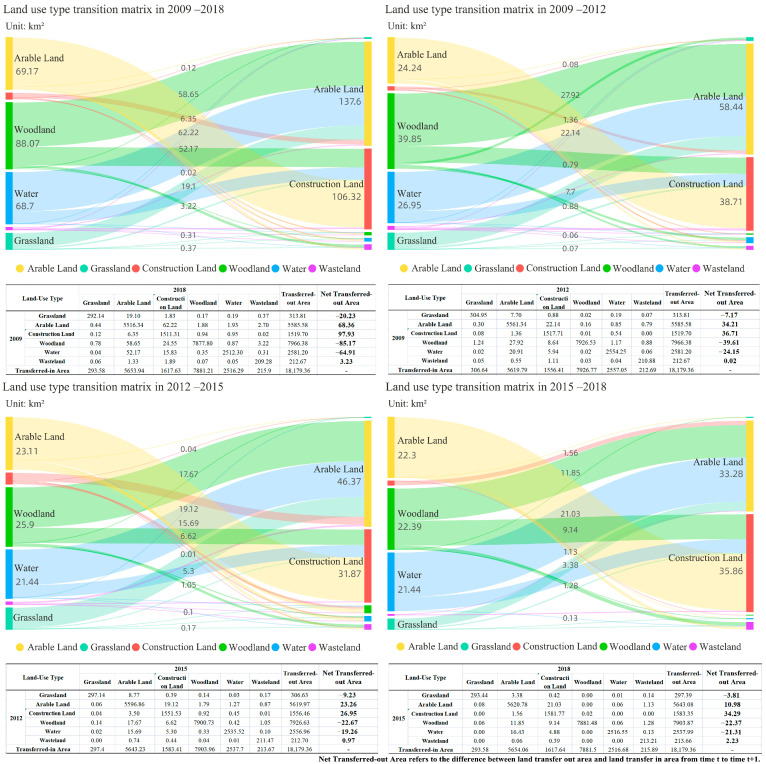
Sankey Diagram of Land Use Type Transfer Matrix of Changde from 2009 to 2018.

**Figure 4 ijerph-19-14289-f004:**
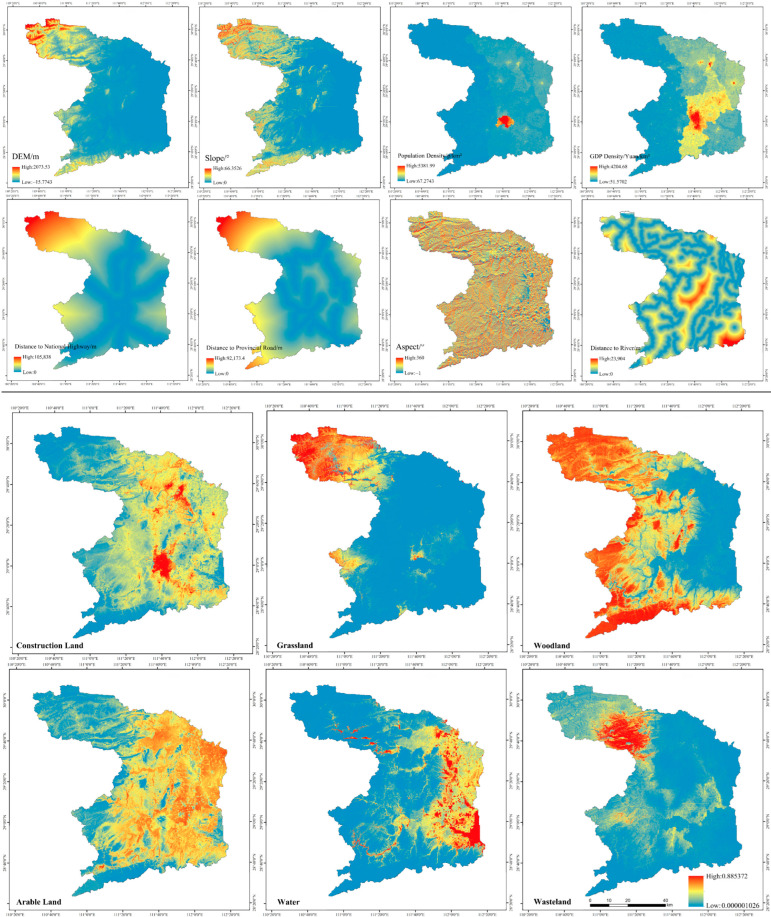
Driving factors for calculating the different land-use probabilities of occurrence in 2018.

**Figure 5 ijerph-19-14289-f005:**
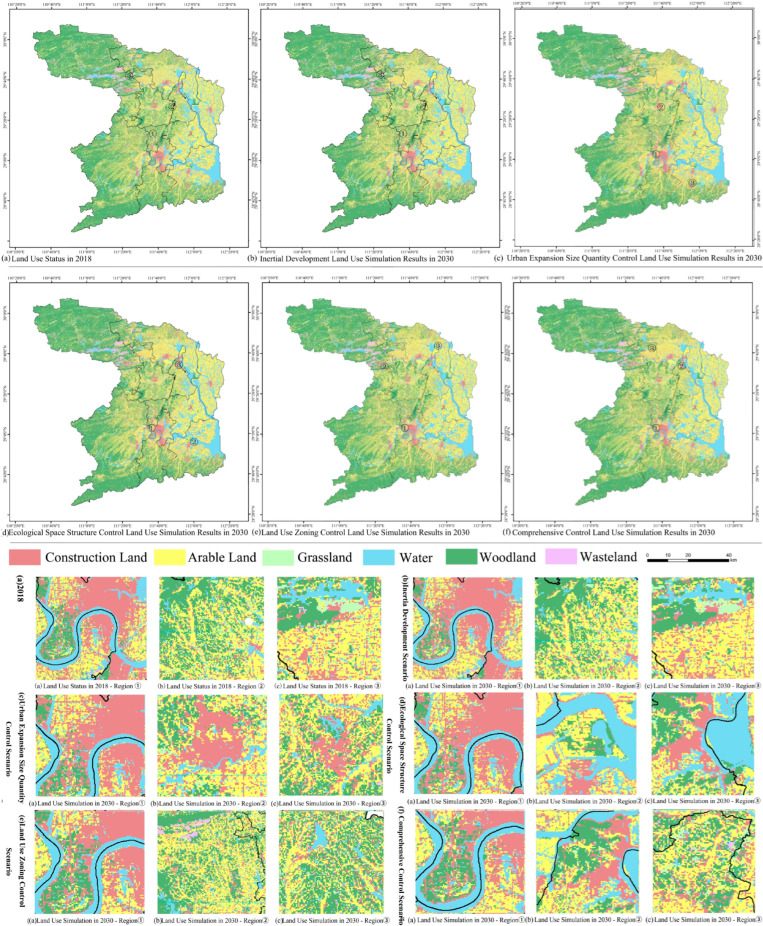
Comparison of the Land Use Status of Actual Changde in 2018 and the Results of the Modeling of Scenarios.

**Figure 6 ijerph-19-14289-f006:**
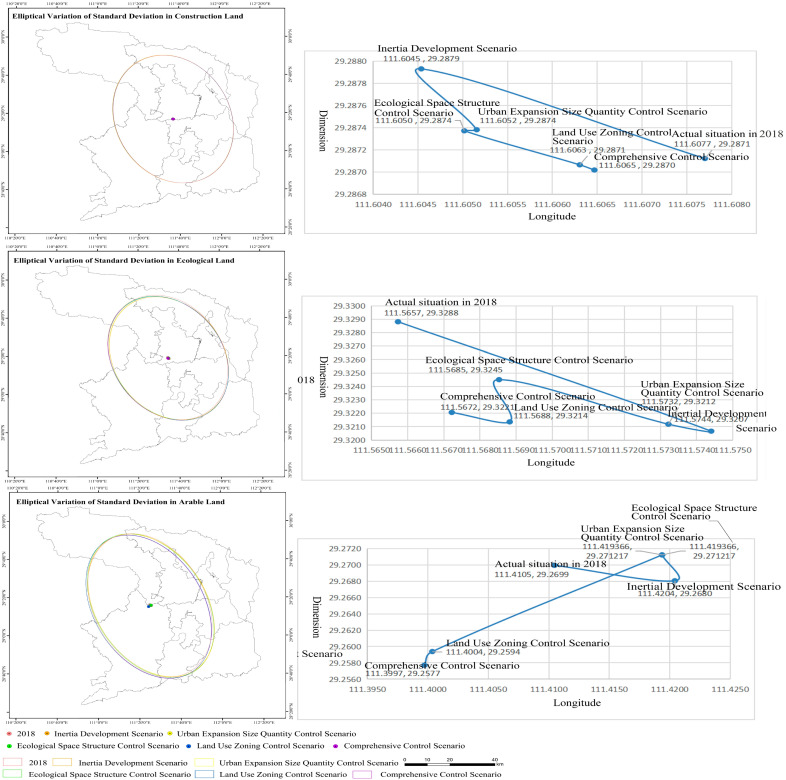
Standard Deviation Ellipse Distribution Map of Land Use Types in Changde City under Different Scenarios.

**Figure 7 ijerph-19-14289-f007:**
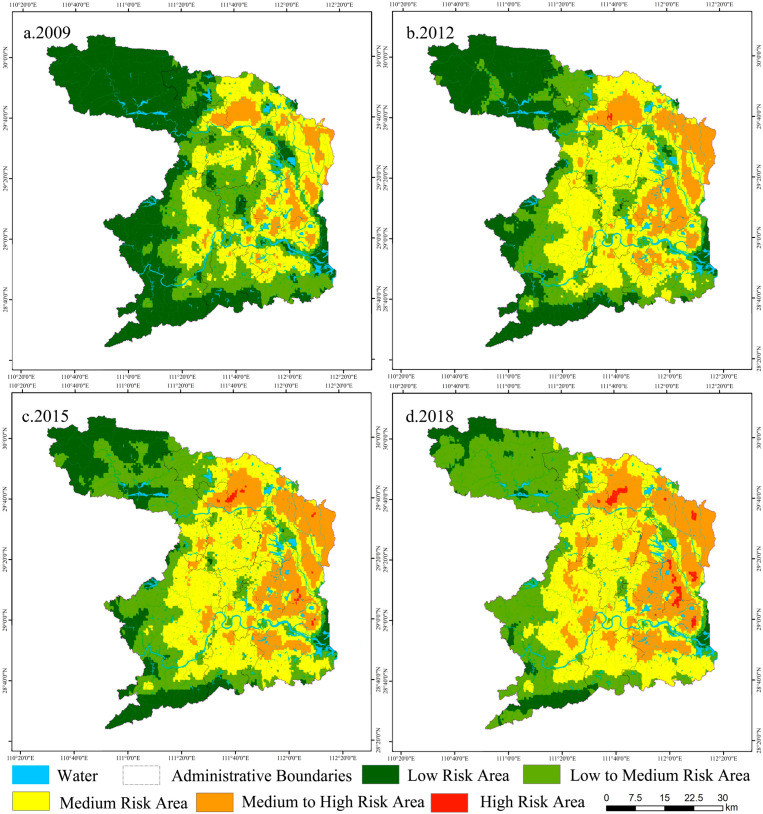
Distribution Map of Landscape Ecological Risk in Changde from 2009 to 2018.

**Figure 8 ijerph-19-14289-f008:**
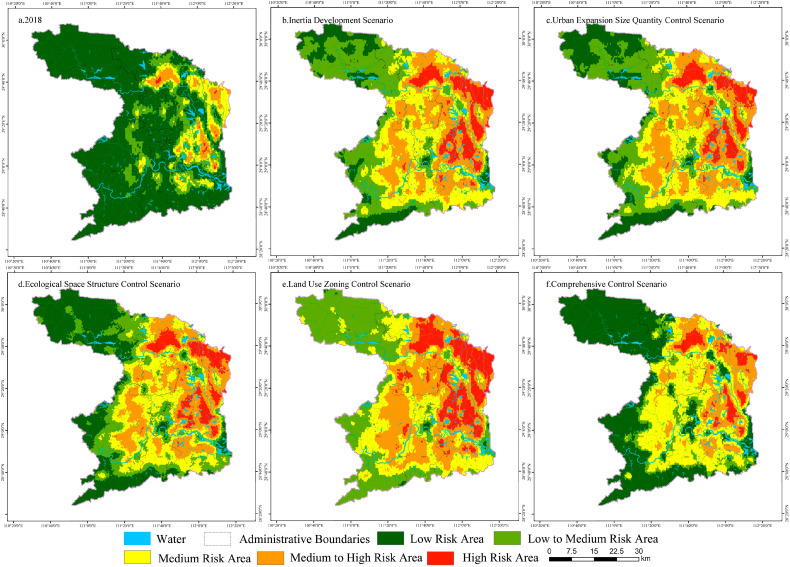
Distribution of Landscape Ecological Risk Levels in Changde under Different Scenarios.

**Figure 9 ijerph-19-14289-f009:**
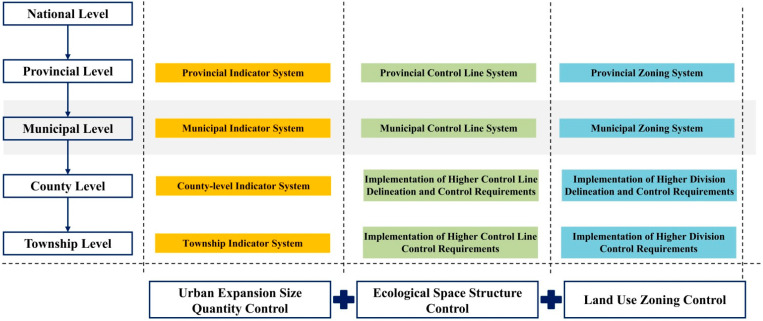
Territorial Spatial Planning Management and Control Transmission System.

**Table 1 ijerph-19-14289-t001:** Summary of Land Use Simulation Models.

Model Type	Model	Analysis Unit	Advantage	Disadvantage	Data Demand
Non-Spatial Simulation Models	Grey-Forecasting Model [31]	Applicable to different scales, with administrative districts as analysis units	The output results are numerical results that facilitate the characterization of mathematical models.	Large amounts of data are needed; it is very sensitive to space-time dependence and heterogeneity; while applicable to investigating processes involved in land-use change, the ability of sample projections is limited.	Large amounts of data are required.
	Logistic Regression Model [32]		The resulting probability parameter favors the assignment of variations according to any scenario.		
	System Dynamics Model [33,34]	Applicable to cities and large areas	It can describe the dynamic mechanism and feedback of complex nonlinear land use systems; it can reduce dependence on data; and it is easy to operate.	It is a lack of simulation of the characteristics of land-use spatial patterns.	Data requirements are small, requiring only land use type-related data and corresponding socio-economic data.
	Artificial Neural Network Model [35]		It has less data dependency; it is easy to operate.	The training model data is demanding, time-consuming, and tedious; there are many unstable factors; and there is no clear conversion rule in the simulation process.	
	Markov Model [36]		The model is relatively simple and flexible; it is able to describe a complex and lengthy land use conversion process regarding simple transition probabilities.	It is a non-spatial simulation; spatial impacts on individual land use types are not taken into account; the effect of adjacent cells is not taken into account.	Data requirements are usually time data sets of continuous time dimension.
Spatial Simulation Models	Cellular Automaton Model [37]	Applicable to different scales, often to model urban growth dynamics	It is one of the simplest LUCC modeling methods; it is directly compatible with raster data and can be implemented in GIS; it has clear conversion rules (relative to ANNs); and it can quickly process data and results.	The model is not estimated using actual data; it is its failure to take into account human decision-making that affects building regional expansion.	It allows for direct processing of land cover data for remote sensing.
	Multibody Model [38]	Suitable for small scale	It can simulate LUCC from different angles by selecting different objects and describing their behaviors according to requirements.	It is less suitable for large-area multi-category land-use change simulations.	Data requirements are a lot of micro and fine data.

**Table 2 ijerph-19-14289-t002:** Data Information and Sources.

Data Type	Data Name	Data Timing	Data Properties	Data Sources	Access Time	Data Usage
Land use	LUCC data	2009, 2012, 2015, 2018	Grids/300 m	Climate Change Initiative-Land Cover (CCI-LC) (http://maps.elie.ucl.ac.be/CCI/viewer/download.php)	30 May 2022	Used for the extraction of land use type information, simulation model input, and simulation accuracy verification of the FLUS model.
Social economy	Population density data	2009	Grids/100 m	WorldPop (https://www.worldpop.org/)	29 May 2022	Used to calculate the suitability probability as an influence factor.
	GDP density data	2009	Grids/1000 m	Global Risk Data Platform (http://preview.grid.unep)	31 May 2022	
	Demographic data	2009–2018	Statistical data	Chang De City Statistical Yearbook	-	
	GDP statistics data	2009–2018	Statistical data	Chang De City Statistical Yearbook	-	
Basic geographic data	Traffic data	2009	Vector	Open Street Map (https://www.openstreetmap.org/)	31 May 2022	Used to calculate the suitability probability as an influence factor.
	Water data	2009	Vector	Calculated from LUCC data	-	
	DEM data	2009	Grids/30 m	NASA SRTM DEM products (http://reverb.echo.nasa.gov/reverb/)	31 May 2022	
	Slope data			Calculated from DEM data	-	
	Aspect data			Calculated from DEM data	-	
Protected areas data	Nature protection area data	2018	Vector	World Database of Protected Areas (https://www.protectedplanet.net/)	31 May 2022	Used to set the restricted area for FLUS model simulation.
Policy documents	Various planning and construction documents of Changde	-	Text	Changde natural resources and Planning Bureau (https://zrzyhghj.changde.gov.cn/)	31 May 2022	Used for setting profile parameters.

**Table 3 ijerph-19-14289-t003:** Landscape Ecological Risk Model.

Name	Landscape Index	Calculation Formula	Ecological Meaning
Landscape Fragmentation Index (*C_i_*)	Patch Density (*PD*)	$PD=\frac{N}{A}$	PD refers to the number of patches per plaque area. It reflects the degree of landscape plaque fragmentation and the shape and size of landscape patches subjected to natural topographic blockade or human construction activity cutting to produce fragmentation of landscape patches. The larger the value (indicating the finer and more fragmented the landscape patch is cut), the higher the degree of landscape fragmentation and the weaker the ability of the characterized landscape to resist risk; where A is the landscape type total area and N is number of landscape patches.
	Inspersion & Juxtapositon Index (*IJI*)	$IJI=\frac{-\sum_{k=1}^{n}\left[\left(\frac{e_{ik}}{\sum_{k=1}^{n}e_{ik}}\right)\ln\left(\frac{e_{ik}}{\sum_{k=1}^{n}e_{ik}}\right)\right]}{\ln\left(m-1\right)}$	IJI can characterize the interspersed and mixed nature of landscape patches. The larger the value, the more complex the plaque dispersal, the higher the degree of fragmentation, and the weaker the landscape’s risk resistance ability; where $e_{ik}$ represents the total length of the edge between landscape patch types *i* and *k*.
Landscape Separation Index (*E_i_*)	Splitting Index (*SPLIT*)	$SPLIT=\frac{A^{2}}{\sum_{i=1}^{m}\sum_{j=1}^{n}a_{ij}^{2}}$	SPLIT characterizes the level of separation between isotype landscape patches The larger the value, the more dispersed and complex the distribution pattern of landscape patches in space, the lower its ecosystem stability, and the weaker the landscape’s risk resistance ability; where A is the total landscape area, m is the number of landscape types, n is the number of landscape patches, and $a_{ij}$ is the area of the jth patch of the ith landscape type.
	Connectivity Index (*CONNECT*)	$CONNECT=\left[\frac{\sum_{i=1}^{m}\sum_{j=1}^{n}c_{ijk}}{\frac{n_{i}\times \left(n_{i}-1\right)}{2}}\right]\times 100$	CONNECT characterize connectivity between landscape patches. The larger the value, the better the accessibility between inner landscape patches and the stronger the landscape’s anti-risk capability; where $n_{i}$ is the number of patches type i, $c_{ijk}$ is the positional relationship between plaque j and k in plaque type i within the threshold distance.
	Patch Cohesion Index (*COHESION*)	$COHESION=\left[1-\frac{\sum_{j=1}^{n}P_{ij}}{\sum_{j=1}^{n}P_{ij}\times \sqrt{a_{ij}}}\right]\times {\left[1-\frac{1}{\sqrt{N}}\right]}^{-1}\times 100$	COHESION is an index that characterizes the degree of agglomeration of a landscape type and it reflects the degree of spatial connectivity between landscape patches. The larger the value, the better the landscape connectivity and the greater the landscape’s anti-risk ability; where $P_{ij}$ is the perimeter of the jth plaque of the ith landscape type, $a_{ij}$ is the area of the jth plaque of the ith landscape type, and N is the total number of landscape cells.
Landscape Dominance Index (*D_i_*)	Largest Patch Index (*LPI*)	$LPI=\frac{\underset{J=1}{\max}\left(a_{ij}\right)}{A}\times 100$	LPI can reflect the dominance of types throughout the landscape, and to some extent, it characterizes the overall nature of human activity versus the magnitude of the weakness; where $a_{ij}$ is the area of the jth patch of the ith landscape type and A is the total landscape area.
	Fractal Dimension Index (*FRAC_MN*)	$FRAC\_MN=\sum_{i=1}^{m}\sum_{j=1}^{n}2\ln\left(\frac{0.25P_{ij}}{\ln A_{ij}}\right)/N$	*FRAC_MN* characterizes the complexity of landscape types. The larger the value, the more complex the landscape plaque shape, the lower the landscape dominance, and the weaker the landscape anti = risk ability. Where $A_{ij}$ is the area of the jth plaque of the ith landscape type and $P_{ij}$ is the perimeter of the jth plaque of the ith landscape type.
Landscape Disturbance Index (*E_i_*)	$E_{i}=a\left(X_{1}\right)+b\left(X_{2}\right)+c\left(X_{3}\right)+d\left(X_{4}\right)+e\left(X_{5}\right)+f\left(X_{6}\right)+g\left(X_{7}\right)$		Represents the degree of disturbance experienced by different landscape types when subjected to external perturbations. The landscape disturbance index is a correlation function of the degree of landscape fragmentation, landscape separation, and landscape dominance; where $X_{1},X_{2},X_{3},X_{4},X_{5},X_{6},X_{7}$ is the landscape pattern index after normalization, and a,b,c,d,e,f,g are the respective weights and summed to 1.
Landscape Vulnerability Index (*F_i_*)	Obtained by expert score assignment	Normalize UVs	Represents the sensitivity of a landscape type i to external perturbants. This study assigns different landscape types using an expert scoring method and performs data normalization based on previous findings.
Landscape Loss Index (*R_i_*)	$R_{i}=\sqrt{E_{i}\times F_{i}}$		Represents the possible loss of ecosystem when a landscape type i is affected by external disturbance factors. It is a function of landscape disturbance and landscape vulnerability.
Landscape Ecological Risk Index (*ERI_k_*)	$ERI_{k}=\overset{n}{\underset{i=1}{\sum }}\frac{A_{ki}}{A_{k}}\times R_{i}$		Represents the overall ecological security level of the regional landscape. The larger the value, the greater the probability that the landscape may suffer from ecological risk and the deeper the degree of impact; where $ERI_{k}$ refers to the landscape ecological risk index of the kth risk unit, $A_{ki}$ represents the area of landscape type i in the kth risk unit; $A_{k}$ represents the area of the kth risk unit; and $R_{i}$ is the landscape loss index.

**Table 4 ijerph-19-14289-t004:** Index Weight.

Index		Weight
Landscape Fragmentation Index	Patch Density (PD)	8.70%
	Inspersion and Juxtapositon Index (IJI)	11.59%
Landscape Separation Index	Splitting Index (SPLIT)	13.18%
	Connectivity Index (CONNECT)	8.50%
	Patch Cohesion Index (COHESION)	13.82%
Landscape Dominance Index	Largest Patch Index (LPI)	14.11%
	Fractal Dimension Index (FRAC_MN)	30.09%

**Table 5 ijerph-19-14289-t005:** The Landscape Vulnerability Index.

Landscape Type	Land Use Type	Landscape Vulnerability Index
Construction Land	Construction Land	0.143
Ecological Land	Woodland	0.857
	Water	
	Grassland	
	Wasteland	
Agricultural Land	Arable Land	0.741

**Table 6 ijerph-19-14289-t006:** Characteristics of Land Use Structure in Changde from 2009 to 2018.

Statistical Type		Grassland	Arable Land	Construction Land	Woodland	Water	Wasteland
2009	Area (km^2^)	313.81	5585.58	1519.70	7966.38	2581.20	212.67
	Proportion (%)	1.73	30.72	8.36	43.82	14.20	1.17
2012	Area (km^2^)	306.64	5619.79	1556.41	7926.77	2557.05	212.69
	Proportion (%)	1.69	30.91	8.56	43.60	14.07	1.17
2015	Area (km^2^)	297.39	5643.08	1583.35	7903.87	2537.99	213.66
	Proportion (%)	1.64	31.04	8.71	43.48	13.96	1.18
2018	Area (km^2^)	293.58	5653.94	1617.63	7881.21	2516.29	215.9
	Proportion (%)	1.61	31.10	8.90	43.35	13.84	1.19
2009–2018	Net variable (km^2^)	−20.23	68.36	97.93	−85.17	−64.91	3.23
	Change rate (%)	−0.12	0.38	0.54	−0.47	−0.36	0.02

**Table 7 ijerph-19-14289-t007:** Characteristics of Land Use Types in Changde under Different Scenarios.

Statistical Indicators		Construction Land	Grassland	Woodland	Arable Land	Water	Wasteland
Area (km^2^)	2018	1617.62	293.58	7881.43	5654.01	2516.66	215.89
	ID Scenario	1844.07	245.87	7709.05	5787.80	2364.70	227.71
	UESQC Scenario	1805.53	246.43	7726.99	5801.35	2370.17	228.73
	ESSC Scenario	1804.54	246.98	7729.26	5787.71	2382.69	228.02
	LUZC Scenario	1737.17	245.45	7709.28	5895.50	2363.67	228.12
	CC Scenario	1721.07	246.79	7717.24	5895.40	2370.99	227.70
Proportion (%)	2018	8.90	1.61	43.35	31.10	13.84	1.19
	ID Scenario	10.14	1.35	42.41	31.84	13.01	1.25
	UESQC Scenario	9.93	1.36	42.50	31.91	13.04	1.26
	ESSC Scenario	9.93	1.36	42.52	31.84	13.11	1.25
	LUZC Scenario	9.56	1.35	42.41	32.43	13.00	1.25
	CC Scenario	9.47	1.36	42.45	32.43	13.04	1.25
Rate of change (%)	ID Scenario	14.00	−16.25	−2.19	2.37	−6.04	5.47
	UESQC Scenario	11.62	−16.06	−1.96	2.61	−5.82	5.95
	ESSC Scenario	11.56	−15.87	−1.93	2.36	−5.32	5.62
	LUZC Scenario	7.39	−16.39	−2.18	4.27	−6.08	5.66
	CC Scenario	6.40	−15.94	−2.08	4.27	−5.79	5.47

ID is an acronym for inertial development. UESQC is an acronym for urban expansion size quantity control. ESSC is an acronym for ecological space structure control. LUZC is an acronym for land use zoning control. CC is an acronym for comprehensive control.

**Table 8 ijerph-19-14289-t008:** Characteristics of Landscape Ecological Risk Index in Changde from 2009 to 2018.

Land Use Type	Year	Landscape Fragmentation Index		Landscape Separation Index			Landscape Dominance Index		Landscape Disturbance Index	Landscape Loss Index	Landscape Ecological Risk Index
		PD	IJI	SPLIT	CONNECT	COHESION	LPI	FRAC_MN			
Construction Land	2009	2.8389	99.7455	97,801.6274	0.0263	71.4872	0.2319	1.5933	0.8433	0.3473	0.0290
	2012	2.8291	99.7002	72,803.094	0.0264	73.662	0.2685	1.592	0.7957	0.3373	0.0289
	2015	2.8235	99.6628	60,378.5114	0.0264	75.1241	0.2858	1.5916	0.7712	0.3321	0.0289
	2018	2.8253	99.6475	43,883.7067	0.0265	77.4236	0.3648	1.5921	0.7416	0.3256	0.0289
Agricultural Land	2009	1.6211	81.7994	283.2084	0.0415	99.0139	3.565	1.6275	0.5378	0.6197	0.1905
	2012	1.6095	82.0783	271.3794	0.0417	99.0368	3.7173	1.6271	0.5361	0.6187	0.1914
	2015	1.6065	82.2704	296.9405	0.0418	98.9963	3.0409	1.6298	0.5500	0.6266	0.1947
	2018	1.6102	82.6283	298.3005	0.0418	98.9912	3.0119	1.6308	0.5563	0.6302	0.1962
Ecological Land	2009	0.9225	78.2172	3.2156	0.0726	99.9456	55.7652	1.5589	0.0003	0.0164	0.0100
	2012	0.943	78.1966	3.2741	0.0714	99.944	55.2646	1.5611	0.0140	0.1093	0.0662
	2015	0.9558	78.1575	3.3228	0.0706	99.9426	54.8586	1.5625	0.0228	0.1397	0.0841
	2018	0.9641	78.4404	3.4041	0.0702	99.94	54.1989	1.5642	0.0342	0.1712	0.1027

**Table 9 ijerph-19-14289-t009:** Changes in Landscape Ecological Risk Levels in Changde from 2009 to 2018.

Landscape Ecological Risk Levels	2009		2012		2015		2018	
	Area (km^2^)	Proportion (%)	Area (km^2^)	Proportion (%)	Area (km^2^)	Proportion (%)	Area (km^2^)	Proportion (%)
Low risk	7410.87	40.77	4907.22	26.99	3245.52	17.85	1183.75	6.51
Low-to-medium risk	4876.33	26.82	4414.03	24.28	4951.51	27.24	6046.14	33.26
Medium risk	4706.72	25.89	6398.84	35.20	6664.74	36.66	6674.42	36.71
Medium-to-high risk	1185.45	6.52	2450.34	13.48	3256.73	17.91	4094.06	22.52
High risk	0.00	0.00	8.94	0.05	60.87	0.33	181.00	1.00

**Table 10 ijerph-19-14289-t010:** Indicators of Landscape Ecological Risk in Changde under Different Scenarios.

Land Use Type	Scenario	Landscape Fragmentation Index		Landscape Separation Index			Landscape Dominance Index		Landscape Disturbance Index	Landscape Loss Index	Landscape Ecological Risk Index
		PD	IJI	SPLIT	CONNECT	COHESION	LPI	FRAC_MN			
Construction Land	ID	2.8178	99.6994	37,836.1552	0.0265	78.5339	0.4069	1.5925	0.7303	0.3232	0.0328
	UESQC	2.6096	98.9137	28,709.8351	0.0272	81.6136	0.4777	1.602	0.7271	0.3224	0.0320
	ESSC	2.6155	98.8757	29,675.6728	0.0271	81.3372	0.4592	1.5974	0.7108	0.3188	0.0317
	LUZC	2.6696	98.9488	38,801.1369	0.0269	79.4144	0.3896	1.5982	0.7384	0.3249	0.0311
	CC	2.6884	98.9677	40,748.9403	0.0268	78.8161	0.3769	1.5986	0.7466	0.3268	0.0309
Agricultural Land	ID	1.5444	84.8313	197.0264	0.0416	99.1485	5.7721	1.628	0.5451	0.6239	0.1986
	UESQC	1.3101	89.3367	131.8138	0.0454	99.3648	6.0577	1.6238	0.5300	0.6152	0.1963
	ESSC	1.3928	89.0239	113.0822	0.0441	99.3764	7.3517	1.6253	0.5380	0.6198	0.1973
	LUZC	1.3828	87.0873	123.828	0.0452	99.3575	6.7449	1.6311	0.5510	0.6272	0.2034
	CC	1.4041	86.8652	137.9975	0.045	99.3352	5.9492	1.6291	0.5452	0.6239	0.2023
Ecological Land	ID	1.0826	81.2016	4.459	0.0607	99.8356	49.4595	1.5789	0.1638	0.3747	0.2174
	UESQC	1.0888	82.6824	4.0934	0.0624	99.9057	49.4199	1.5811	0.1602	0.3705	0.2155
	ESSC	1.0677	82.1896	3.9473	0.0627	99.9139	50.3273	1.5792	0.1457	0.3533	0.2058
	LUZC	1.0841	80.2014	7.3444	0.0634	99.8557	31.8315	1.5777	0.1765	0.3889	0.2256
	CC	1.0681	80.0156	4.0293	0.0634	99.9153	49.8109	1.5727	0.1073	0.3033	0.1762

## Data Availability

Not applicable.

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
