# Peer review of "Multi-Scenario Simulation of Land Use and Landscape Ecological Risk Response Based on Planning Control"

_ijerph, 2022, doi:10.3390/ijerph192114289_

Round 1

Reviewer 1 Report

The manuscript addresses a relevant problem in spatial planning and its authors propose an articulated set of tools to simulate scenarios of LU change and asses the related ecological risk. The findings are well presented and backed up by the supplied tables and figures; nonetheless, the methods are introduced only briefly (to keep the length of the manuscript) but in a few cases without enough clarity and proper references to first-hand sources that would help the interested reader. It is appreciable that the authors transparently highlighted the limitations of the present study in the discussion section.

The following are my remarks, mainly focused on the Methods section (3). The numbers refer to lines in the manuscript.

l. 121 Table 1 suffers from poor readability: lack of horizontal alignment of some cells hinders the great utility of such a synopsis. I have found it hard to interpret the exact correspondence across columns, and in the end, I am afraid some of the cells include only partial text. Authors should try to simplify the contents.

l.145 Table 2: Separate the information about the access time from the URLs  consistently across the rows.

l.157-158 Authors would clarify how they have used the data observed with regard to other years (e.g., 2012, 2015, 2018 for Land Use, clearly needed to estimate the transitions).

l.162-164 In my opinion, this short sentence does not help to understand what the ANN modelling is used for; it would be worthwhile to explain that ANN is a computational model able to learn some general patterns related to a complex system once trained by means of a sufficiently large set of observations about its dynamics. Being a complex model itself, I see that it is quite hard to give a brief introduction to it, so I would suggest authors focus on the specific way the ANN tool is exploited in the FLUS model: learn the transition probabilities for the cells that will be used as parameters in the Cellular Automata simulation of Land Use change.

l.169 and following: Bearing on my previous remark, I would discourage including maths about ANN that would be never used; the reference to the original paper about FLUS [34], complemented by an introductory one [e.g., Li, X., & Yeh, A.G. (2002). Neural-network-based cellular automata for simulating multiple land use changes using GIS. International Journal of Geographical Information Science, 16, 323 - 343, DOI:10.1080/13658810210137004] would suffice; moreover, a sketch of the network architecture would be helpful to undertand input and outputs, being at the same time highly more informative for the readers. If authors are of different advice, I should stress that they are committed to properly introducing and explaining any symbol used, being consistent in the use of indexes (to not confuse readers) across all the subsections. In the present form, the math is quite hard to follow, it seems to add little to the discourse, and is poorly presented –  f.i., at l. 202 P is capitalized, and this does not help, added to the font-face differences  (symbols included in the equations do not appear consistently italicised in the explaining text).

l.210 precision are (missing 'a')...

l.211 are (missing 'a') to the spatial pattern...

l.215 please, specify that r=6 is the number of distinct LU categories considered to assemble the confusion matrix.  

l.233 In Table3, the formula related to Inspersion & Juxtapositon Index (IJI) includes an undefined symbol: eik: please, clarify if k is an index and EI is the Inspersion & Juxtapositon Index (IJI) recalled in the following.

l.237 It would be beneficial to cite the supporting literature for the procedure adopted, not simply the commercial software (SPSS Pro) nor the specific cloud instance used (SPSS AU);
 e.g., Diakoulaki, D., Mavrotas, G., & Papayannakis, L. (1995). Determining objective weights in multiple criteria problems: The critic method. Computers & Operations Research, 22(7), 763–770. doi: 10.1016/0305-0548(94)00059-H

l.271 In Figure 2, it would be more immediate presenting in the marginal of tables (Transferred-in and Transferred-out areas) as the net values of changes (in and out) as reported in the Sankey diagrams (simply subtracting the values of the area that did not change LU category), also to easily follow the remarks at lines 265-268.

l.251 In the subsection title lower case a:  Analysis of Land Use Evolution *A*and Analog Verification

l.275 It would be beneficial to the reader to understand which years have actually been used to train the ANN (since the network is supposed to learn the transition pattern at least two different years are involved (e.g., see Fig. 5 IN NBaig, M. F., Mustafa, M. R. U., Baig, I., Takaijudin, H. B., & Zeshan, M. T. (2022). Assessment of Land Use Land Cover Changes and Future Predictions Using CA-ANN Simulation for Selangor, Malaysia. Water, 14(3), 402. DOI: 10.3390/w14030402), so the 'actual LU situation' could relate only to an assessment of the prediction, that is in the aftermath of the CA simulation.

l.304 This is the first time the simulation model is recalled as a Markov Model; I think authors have used the CA-Markov model, so far introduced as 'Cellular Automaton" in Table 1 and in a footnote to it. Here I can see two needs for clarification: (i) that this part of the model application coincides with the CA introduced in the Methods section – if I understand it correctly, and (ii) that the transition probabilities of the Markov model have been calibrated based on the changes observed between a base year and a second one (or more than just one other year, I should admit this point remains unclear to me so far).
Overall, I would warmly suggest normalizing the language and sticking with Cellular Automata, since each cell in the simulation grid will act as a cellular automaton to make the whole system path of LUC emerge.

Author Response

Response to Reviewer 1 Comments

On behalf of all the contributing authors, I would like to express our sincere appreciations of your constructive comments concerning our article entitled “Multi-Scenario Simulation of Land Use and Landscape Eco-logical Risk Response Based on Planning Control” (ID:ijerph-1970251). These comments are all valuable and helpful for improving our article. According to your comments, we have made extensive modifications to our manuscript. The main corrections are in the manuscript and the responds to your comments are as follows:

Point 1:. 121 Table 1 suffers from poor readability: lack of horizontal alignment of some cells hinders the great utility of such a synopsis. I have found it hard to interpret the exact correspondence across columns, and in the end, I am afraid some of the cells include only partial text. Authors should try to simplify the contents.

Response 1: Your suggestion really means a lot to us. Yes, if we simplify the content of Table 1 and add a horizontal line in the table, it will be easier for readers to understand. Therefore, in the revised version, we have simplified the advantages and disadvantages of various models and methods in the table, and added horizontal lines in the table, which can help readers get the information in the table more quickly. We marked the revised part in red font.

Point 2: 145 Table 2: Separate the information about the access time from the URLs consistently across the rows.

Response 2: We think this is an excellent suggestion. We have added the column of access time in Table 2, and we have supplemented the specific use of each data in this study, so as to help readers better understand the data.

Point 3: 157-158 Authors would clarify how they have used the data observed with regard to other years (e.g., 2012, 2015, 2018 for Land Use, clearly needed to estimate the transitions).

Response 3: We agree with the reviewer’s assessment. We have added the use of data in Table 2, so as to supplement the use of land use data in 2009, 2012, 2015 and 2018. In line165-172and line190-202, we made it clear that we applied the land use data in 2009 and 2018 and the driving factor data in 2009 and 2018 to calculate the suitability probability of land use types; And the Markov transfer probability matrix is calculated by using the land use data of 2009, 2015 and 2018. The data in 2009, 2012, 2015 and 2018 you mentioned are the continuous time data we need to analyze the characteristics of land use evolution, and also the data we need to assess the time evolution characteristics of landscape ecological risks.

Point 4: 162-164 In my opinion, this short sentence does not help to understand what the ANN modelling is used for; it would be worthwhile to explain that ANN is a computational model able to learn some general patterns related to a complex system once trained by means of a sufficiently large set of observations about its dynamics. Being a complex model itself, I see that it is quite hard to give a brief introduction to it, so I would suggest authors focus on the specific way the ANN tool is exploited in the FLUS model: learn the transition probabilities for the cells that will be used as parameters in the Cellular Automata simulation of Land Use change.

Response 4: We think this is an excellent suggestion. Because the explanation of the principle of the ANN model itself is not meaningful in the study, we have emphasized the role of the ANN model in the FLUS model in line204-214, which is to obtain the suitability probability of land use types for the FLUS model, and added references [43] to the application of the FLUS model in line214 to ensure the scientific rationality of our application of the FLUS model.

Point 5: 169 and following: Bearing on my previous remark, I would discourage including maths about ANN that would be never used; the reference to the original paper about FLUS [34], complemented by an introductory one [e.g., Li, X., & Yeh, A.G. (2002). Neural-network-based cellular automata for simulating multiple land use changes using GIS. International Journal of Geographical Information Science, 16, 323 - 343, DOI:10.1080/13658810210137004] would suffice; moreover, a sketch of the network architecture would be helpful to undertand input and outputs, being at the same time highly more informative for the readers. If authors are of different advice, I should stress that they are committed to properly introducing and explaining any symbol used, being consistent in the use of indexes (to not confuse readers) across all the subsections. In the present form, the math is quite hard to follow, it seems to add little to the discourse, and is poorly presented –  f.i., at l. 202 P is capitalized, and this does not help, added to the font-face differences  (symbols included in the equations do not appear consistently italicised in the explaining text).

Response 5: We agree with the reviewer’s assessment. We have shifted the interpretation focus of the ANN model to the specific role in the FLUS model in line178-188. In order to better understand the relationship between inputs and outputs in the FLUS model, we have referred to [34] and [43]. We have supplemented the discussion of the framework in line 145-149, and added the framework (Figure 2) in line 150. (Liu, X. P.; Liang, X.; Li, X.; Xu, X. C.; Ou, J. P.; Chen, Y. M.; Li, S. Y.; Wang, S. J.; Pei, F  S., A future land use simulation model (FLUS) for simulating multiple land use scenarios by coupling human and natural effects.  LANDSCAPE AND URBAN PLANNING 2017, 168, 94-116.) (Li, X.; Yeh, A., Neural Network Based Cellular Automata for Simulating Multiple Land Use Changes Using GIS. International Journal of Geographic Information Science 2002, 16, 323-343.).

At the same time, we have revised the formula symbols in the manuscript to ensure that all formulas are expressed in italics.

Point 6: 210 precision are (missing 'a')...

Response 6: It is really a giant mistake to the whole quality of our article. We feel sorry for our carelessness. We have corrected it and we also feel great thanks for your point out. As suggested by the reviewer, we have changed "re" to "are".

Point 7: 211 are (missing 'a') to the spatial pattern...

Response 7: As suggested by the reviewer, We have changed "re" to "are". And we have checked and revised the grammar and spelling in the whole manuscript.

Point 8: 215 please, specify that r=6 is the number of distinct LU categories considered to assemble the confusion matrix. 

Response 8: We agree with the reviewer’s assessment. We have added in line260-262 that r is the number of distinct LU categories considered to assemble the fusion matrix. In our research, we have six LU categories, so we set r to 6.

Point 9: 233 In Table3, the formula related to Inspersion & Juxtapositon Index (IJI) includes an undefined symbol: eik: please, clarify if k is an index and EI is the Inspersion & Juxtapositon Index (IJI) recalled in the following.

Response 9: We agree with the reviewer’s assessment. We have supplemented  in the Penetration & Juxtaposition Index (IJI). Here  represents the total length of the edge between landscape patch types i and k. The smaller the IJI value is, the more balanced the patch distribution is, the stronger the anti risk ability of the landscape is.

Point 10: 237 It would be beneficial to cite the supporting literature for the procedure adopted, not simply the commercial software (SPSS Pro) nor the specific cloud instance used (SPSS AU);

 e.g., Diakoulaki, D., Mavrotas, G., & Papayannakis, L. (1995). Determining objective weights in multiple criteria problems: The critic method. Computers & Operations Research, 22(7), 763–770. doi: 10.1016/0305-0548(94)00059-H

Response 10: As suggested by the commentator, we have added more references to support our -use of CRITIC weighting in line 289-291. According to literature [43] (Diakoulaki, D.;  Mavrotas, G.; Papayannakis, L., Determining objective weights in multiple criteria problems: The CRITIC method. Computers & OR 1995, 22, 763-770.), CRITIC weighting method can reduce the subjective characteristics in the decision-making process by combining subjective weights and objective weights, and can be applied to the pre decision-making stage to more objectively measure the weights of various indicators in this study.

Point 11: 271 In Figure 2, it would be more immediate presenting in the marginal of tables (Transferred-in and Transferred-out areas) as the net values of changes (in and out) as reported in the Sankey diagrams (simply subtracting the values of the area that did not change LU category), also to easily follow the remarks at line 265-268.

Response 12: As suggested by the reviewer, we have added a column of net change area data to the table below Sankey Chart in Figure 3 and highlighted it, so as to more intuitively show the transfer profile of land use types in different periods, and help readers better understand the transfer characteristics between land use types.

Point 13: 251 In the subsection title lower case a:  Analysis of Land Use Evolution *A*and Analog Verification

Response 12: It is really a giant mistake to the whole quality of our article. We feel sorry for our carelessness. We have corrected it and we also feel great thanks for your point out. As suggested by the reviewer, we have changed "A" to "a", and revised the capitalization in the manuscript.

Point 13: 275. It would be beneficial to the reader to understand which years have actually been used to train the ANN (since the network is supposed to learn the transition pattern at least two different years are involved (e.g., see Fig. 5 IN NBaig, M. F., Mustafa, M. R. U., Baig, I., Takaijudin, H. B., & Zeshan, M. T. (2022). Assessment of Land Use Land Cover Changes and Future Predictions Using CA-ANN Simulation for Selangor, Malaysia. Water, 14(3), 402. DOI: 10.3390/w14030402), so the 'actual LU situation' could relate only to an assessment of the prediction, that is in the aftermath of the CA simulation.

Response 13: We agree with the reviewer’s assessment. Therefore, we emphasized the application of ANN model in line191-196. The application of ANN model in this study includes two aspects. The first is to verify that the ANN model can explain land use change simulation accurately enough. Second, calculate the 2018 land use suitability probability and combine it with land use data to simulate the land use change in 2030. However, considering the length of the manuscript, we did not mention the verification of the ANN model in the manuscript.

Point 14: 304 This is the first time the simulation model is recalled as a Markov Model; I think authors have used the CA-Markov model, so far introduced as 'Cellular Automaton" in Table 1 and in a footnote to it. Here I can see two needs for clarification: (i) that this part of the model application coincides with the CA introduced in the Methods section – if I understand it correctly, and (ii) that the transition probabilities of the Markov model have been calibrated based on the changes observed between a base year and a second one (or more than just one other year, I should admit this point remains unclear to me so far).

Response 14: We agree with the reviewer’s assessment. We have emphasized the data use and the method application of Markov Model in the research in line165-173. The application of Markov Model in this study mainly includes two aspects. First, in order to verify that the accuracy of FLUS model in simulating land use change meets the requirements, it is necessary to calculate the land use transfer probability matrix based on the land use data in 2009 and 2015, so as to simulate and predict the land use change in 2018, and verify the accuracy with the actual situation in 2018 ; second, when simulating the land use change under different scenarios in 2030, we adjusted the probability based on the 2009 and 2018 land use transfer probability matrices according to different scenario rule settings, so as to obtain different land use demand scales according to the different scenarios.

Point 15: I would warmly suggest normalizing the language and sticking with Cellular Automata, since each cell in the simulation grid will act as a cellular automaton to make the whole system path of LUC emerge.

Response 15: Thank you for pointing this out. Although we agree with CA model's ability to depict LU system changes, it is not suitable for inclusion in this manuscript. The CA model is a built-in part of the FLUS model. According to the model framework in reference [34] ( Liu, X. P.;  Liang, X.;  Li, X.;  Xu, X. C.;  Ou, J. P.;  Chen, Y. M.;  Li, S. Y.;  Wang, S. J.; Pei, F. S., A future land use simulation model (FLUS) for simulating multiple land use scenarios by coupling human and natural effects. LANDSCAPE AND URBAN PLANNING 2017, 168, 94-116.) , the FLUS model is an integrated model of CA model and SD model. However, the use of the FLUS model is divided into two parts, one is the ANN based Reliability Probability Estimation, and the other is the Land Use Scenario Simulation Based on Self Adaptive Inertia and Competition Mechanism CA.

Reviewer 2 Report

The submitted manuscript concerns the important issue of the multi-scenario simulation of land use and landscape ecological risk response based on planning control. The study applied territorial spatial planning control to a land use multi-scenario simulation in Changde, China, and measured landscape ecological risk response. It embedded five planning control schemes respectively involving inertial development, urban expansion size quantity control, ecological spatial structure control, land use zoning control, and comprehensive control. Remarks: Lines 582: The choice of reference should be supplemented with respect to the future reseach concerning the systematic quantitative research on landscape ecological risk assessment cannot be carried out comprehensively considering all factors, due to the diversity, complexity and nonlinearity of ecosystem dynamics. With the deepening of the understanding of the regional ecosystem in the future, a more scientific evaluation model would be constructed to evaluate the landscape ecological risk more systematically and comprehensively,  e.g. Ref. Modelling water distribution network failures and deterioration, 2017, IEEE International Conference on Industrial Engineering and Engineering Management 2017-December, pp. 924-928, DOI 10.1109/IEEM.2017.8290027, e.g. Ref. Failure Risk Analysis of Water Distributions Systems Using Hydraulic Models on Real Field Data. Ekon. Åšr. 2019, 1, 152–165. https://doi.org/10.34659/d0cp-hn27. If possible, please paste the bigger Figure 5. Standard Deviation Ellipse Distribution Map of Land Use Types in Changde City Under Different Scenarios (line 426). What definition of the ecological risk  do you propose? In the paper the overall landscape ecological risk index expanded over 2009–2018, presenting and S-type time evolution curve of “sharp increase–mitigation” was taken into account. Did you consider others method of ordinary kriging method in the geostatistical method module 458 ArcGIS 10.2? What about the last four years of observation? Do you anticipate obtaining the similar results? Line 460: Give some more information about assumed values of risk, which was divided into low-risk area (ERI≤0.2), medium-to-low-risk area (0.2<eri≤0.3), medium-risk="" area="" 461="" (0.3=""><eri≤0.4), medium-to-high="" risk="" area="" (0.4=""><eri≤0.5), and="" high-risk="" area="" (eri="">-0.5), etc. Give some basis for these categories. </eri≤0.5),></eri≤0.4),></eri≤0.3),>

Author Response

Response to Reviewer 2 Comments

On behalf of all the contributing authors, I would like to express our sincere appreciations of your  constructive comments concerning our article entitled “Multi-Scenario Simulation of Land Use and Landscape Eco-logical Risk Response Based on Planning Control” (ID:ijerph-1970251). These comments are all valuable and helpful for improving our article. According to your comments, we have made extensive modifications to our manuscript. The main corrections are in the manuscript and the responds to your comments are as follows .

Point 1:. Lines 582: The choice of reference should be supplemented with respect to the future research concerning the systematic quantitative research on landscape ecological risk assessment cannot be carried out comprehensively considering all factors, due to the diversity, complexity and nonlinearity of ecosystem dynamics. With the deepening of the understanding of the regional ecosystem in the future, a more scientific evaluation model would be constructed to evaluate the landscape ecological risk more systematically and comprehensively,  e.g. Ref. Modelling water distribution network failures and deterioration, 2017, IEEE International Conference on Industrial Engineering and Engineering Management 2017-December, pp. 924-928, DOI 10.1109/IEEM.2017.8290027, e.g. Ref. Failure Risk Analysis of Water Distributions Systems Using Hydraulic Models on Real Field Data. Ekon. Åšr. 2019, 1, 152–165. https://doi.org/10.34659/d0cp-hn27.

Response 1: Your suggestion is really important to us. Therefore, we have supplemented the demand of current research for comprehensive assessment of landscape ecological risk in the revised version, and supplemented references [46] [47] (Han, N.;  Yu, M.; Jia, P., Multi-Scenario Landscape Ecological Risk Simulation for Sustainable Development Goals: A Case Study on the Central Mountainous Area of Hainan Island. International Journal of Environmental Research and Public Health 2022, 19, 4030. ) (Luo, F.;  Liu, Y.;  Peng, J.; Wu, J., Assessing urban landscape ecological risk through an adaptive cycle framework. Landscape and Urban Planning 2018, 180, 125-134.) in line636-637. In the revised version, we marked the revised part in blue font.

Point 2: If possible, please paste the bigger Figure 5. Standard Deviation Ellipse Distribution Map of Land Use Types in Changde City Under Different Scenarios (line 426).

Response 2: We think this is an excellent suggestion. Therefore, we have revised the layout of Figure 6 to make the picture information more clearly presented to the readers.

Point 3: What definition of the ecological risk do you propose?

Response 3: We have supplemented the definition of ecological risk in line266-271, and analyzed the relationship between ecological risk and landscape ecological risk, so as to help readers better understand the assessment of landscape ecological risk. Ecological risk refers to the potential and potential hazard degree of adverse impact on the assessment endpoint caused by risk factors such as environmental pollution, human stress or natural disasters in different types of ecosystems on a regional scale. In this study, in order to better describe the ecological risk caused by land use and cover change, we narrowed the concept to landscape ecological risk, and built a landscape ecological risk assessment model.

Point 4:. In the paper the overall landscape ecological risk index expanded over 2009–2018, presenting and S-type time evolution curve of “sharp increase–mitigation” was taken into account. Did you consider others method of ordinary kriging method in the geostatistical method module 458 ArcGIS 10.2? What about the last four years of observation? Do you anticipate obtaining the similar results?

Response 4: As suggested by the referee, we have tried our best to verify the applicability of other Kriging methods in this study. However, when using the Pan Kriging method, we found that the spatial and temporal differentiation characteristics of landscape ecological risks in the results were not significant. The change of risk level in space is not obvious, and it is difficult to obtain similar results. The Kriging method includes the general Kriging method and the pan Kriging method. Among them, the common kriging method is the most common and widely used kriging method, which is a default method. The pan kriging method assumes that there is a coverage trend in the data. We can only use this method when we understand that there is a trend in the data and can provide scientific judgment to describe the pan kriging method. However, our understanding of the trend of landscape ecological risk results is not comprehensive, so we use the ordinary Kriging method to calculate.

Point 5:. Line 460: Give some more information about assumed values of risk, which was divided into low-risk area (ERI≤0.2), medium-to-low-risk area (0.2<eri≤0.3), medium-risk="" area="" 461="" (0.3=""><eri≤0.4), medium-to-high="" risk="" area="" (0.4=""><eri≤0.5), and="" high-risk="" area="" (eri="">-0.5), etc. Give some basis for these categories. </eri≤0.5),></eri≤0.4),></eri≤0.3),>

Response 5: We think this is an excellent suggestion. Therefore, we have supplemented the basis for delineation of landscape ecological risk in line516-519. We mainly used the natural mutation point method and the relative index method to divide the ecological risk index value of each subarea into five grades. We have also referred to some literatures in grading, so we have also supplemented literature [44] [45]in line520 ( Cui, L.;  Zhao, Y.;  Liu, J.;  Han, L.;  Ao, Y.; Yin, S., Landscape ecological risk assessment in Qinling Mountain. Geological Journal 2018, 53.). (Xu, Q.;  Guo, P.;  Jin, M.; Qi, J., Multi-scenario landscape ecological risk assessment based on Markov–FLUS composite model. Geomatics, Natural Hazards and Risk 2021, 12, 1448-1465.)
